# Functional analysis of recurrent *CDC20* promoter variants in human melanoma

Paula M. Godoy[1], Abimbola Oyedeji [2,3], Jacqueline L. Mudd [2,3], Vasilios A. Morikis[4], Anna P. Zarov [1], Gregory D. Longmore [3,4], Ryan C. Fields [2,3] & Charles K. Kaufman [1,3✉]

Small nucleotide variants in non-coding regions of the genome can alter transcriptional regulation, leading to changes in gene expression which can activate oncogenic gene regulatory networks. Melanoma is heavily burdened by non-coding variants, representing over 99% of total genetic variation, including the well-characterized TERT promoter mutation. However, the compendium of regulatory non-coding variants is likely still functionally under-characterized. We developed a pipeline to identify hotspots, i.e. recurrently mutated regions, in melanoma containing putatively functional non-coding somatic variants that are located within predicted melanoma-specific regulatory regions. We identified hundreds of statistically significant hotspots, including the hotspot containing the TERT promoter variants, and focused on a hotspot in the promoter of CDC20. We found that variants in the promoter of CDC20, which putatively disrupt an ETS motif, lead to lower transcriptional activity in reporter assays. Using CRISPR/Cas9, we generated an indel in the CDC20 promoter in human A375 melanoma cell lines and observed decreased expression of *CDC20*, changes in migration capabilities, increased growth of xenografts, and an altered transcriptional state previously associated with a more proliferative and less migratory state. Overall, our analysis prioritized several recurrent functional non-coding variants that, through downregulation of *CDC20*, led to perturbation of key melanoma phenotypes.

[1] Division of Medical Oncology, Department of Medicine and Department of Developmental Biology, Washington University School of Medicine, St. Louis, MO, USA. [2] Department of Surgery, Washington University School of Medicine, St. Louis, MO, USA. [3] Siteman Cancer Center, Washington University in Saint Louis, St. Louis, MO, USA. [4] Departments of Medicine (Oncology) and Cell Biology and Physiology and the ICCE Institute, Washington University School of Medicine, St. Louis, MO 63110, USA. ✉email: ckkaufman@wustl.edu

With the widespread availability of whole-genome sequencing and fewer discoveries of novel functional coding mutations, recent efforts have increasingly focused on identification and characterization of variants in the non-coding space of cancer genomes. Cis-regulatory variants (CRV) modulate transcription by altering the regulatory landscape of a gene, which in turn can lead to dysregulation of genes involved in cancer-driving pathways. Identifying CRVs of interest is therefore, generally, a three-step process: (1) identification of variants by whole-genome or targeted sequencing, (2) validation of variants through reporter assays and/or precise genome editing, (3) and characterization of the effect of the gene targeted by the CRV on tumorigenesis or cancer cell biology. For example, TERT promoter mutations were one of the earliest highly recurrent non-coding mutations identified in melanoma and are remarkable due to both a strong activating effect and prevalence in multiple cancers[1–3]. Present in ~80% of cutaneous melanomas, the TERT promoter mutation creates a novel ETS motif that leads to binding of GABPA and de-repression of TERT[4]. The full extent of TERT's influence on tumorigenesis, particularly via this regulatory variant, is still emerging, including its canonical role on telomere maintenance[3,5]. Beyond TERT promoter variants, few other CRVs have been identified and characterized in melanoma[1,2,6–10]. The next most common mutations in cutaneous melanoma are coding mutations in the MAPK pathway, predominantly $BRAF^{V600E/K}$ and $NRAS^{Q61K}$, as well as loss of key tumor suppressors like TP53, PTEN, and CDKN2A[11,12], all with relatively clear canonical growth regulatory and proliferative functions.

Taking a more global view of gene expression, numerous RNA-sequencing studies, both from bulk and single-cell sources, have detected distinct transcriptional states in various melanoma populations[13–16]. For example, high levels of *MITF* are associated with a more proliferative/melanocytic state, while high levels of *AXL* are associated with increased invasive and drug resistant capabilities[13,15,17]. Additionally, aspects of a neural crest transcriptional program, present in the developmental precursors of melanocytes, are frequently identified as distinct subpopulations within tumors and are prominent in the first cells of melanoma[13,15,18–22]. Aside from amplifications of MITF in 5-10% of melanomas, no other recurrent protein coding mutations have been associated with these distinct transcriptional subpopulations[11,12], leading to our hypothesis that CRVs could be a source of transcriptional dysregulation.

Guided by the threefold process described above, we leveraged whole genome sequencing of 183 melanomas from the International Cancer Genome Consortium and 69 melanoma-specific chromatin functional datasets to identify recurrent non-coding variants enriched in potentially functional enhancers/promoters. We validated several variants in the CDC20 promoter which decrease *CDC20* promoter/enhancer-dependent reporter gene expression. We went further to genome engineer a small (~10 bp) and large (~90 bp) promoter indel using CRISPR/Cas9 at this variant location in melanoma cell lines, which leads to decreased *CDC20* expression, and then characterize the potential effects of the variant on in vitro and in vivo cell viability, migration, and global gene expression changes.

## Results

### Putative regulatory regions in melanoma are enriched for hotspot mutations.

To identify recurrent non-coding mutations in human melanoma, we used variants called from whole genome sequencing (WGS) data from the International Cancer Genome Consortium (ICGC), the largest collection of WGS for melanoma to our knowledge, including 183 melanoma samples made up of 75 primary tumors, 93 metastases, and 15 human melanoma cell lines, as exome sequencing does not include full promoters or distal regulatory elements. The bulk of these tumors are cutaneous (140) but includes 35 acral and 8 mucosal melanomas. A total of 20,894,255 substitutions and 96,467 indels were identified from the ICGC Melanoma cohort[12].

To refine our search space, we collated 69 previously published ChIP-seq and ATAC-seq datasets that were specifically performed on melanoma or melanocyte samples[18,23–41] (Supplemental Data 1). We reasoned these regions of the genome are more likely to bind transcription/chromatin factors and refer to them as putative melanoma regulatory regions (pMRRs), thousands of which have been functionally validated in a previous report[42]. Genomic regions outside the pMRRs (red box, indicated by the lack of peak, Fig. 1a) serve as an empirical null distribution but still have large numbers of recurrent mutations.

pMRRs account for only ~12% of the genome and harbor 2,142,063 variants (~10% of total variants detected in the ICGC cohort). Of these, 444,161 variants are merged into 118,741 hotspots (3 or more variants within 25 bp are merged). Our empirical null distribution accounts for 5,478,131 variants within 1,462,992 hotspots. The remaining variants are isolated (i.e. not within 25 bp of another variant) and thus were not designated as hotspots.

All hotspots are also scored based on recurrence (donor score) and the average predicted impact of all variants within a hotspot as computed by the FunSeq2 algorithm, which weighs attributes such as evolutionary conservation and likelihood of TF motif creation/destruction (Funseq2 score, Fig. 1a')[43]. Hotspots in pMRRs have higher hotspot scores (product of donor score and FunSeq2 score) than those in null regions (Fig. 1b). While donor scores are 4.9-fold higher in hotspots within pMRRs than those in null regions, FunSeq2 scores are 6.7-fold higher, drastically reducing the hotspot scores in regions outside of pMRRs and therefore potentially reducing false positives (Fig. 1c).

Promoter regions are enriched in statistically significant test hotspots, while top-scoring null hotspots are commonly found in intergenic regions (Fig. 1d). We identified 140 hotspots with FDR-adjusted p-values $< 2.2 \times 10^{-16}$ encompassing 2,631 mutations, notably including the known TERT promoter variant which has the 13th highest hotspot score (Supplemental Data 2, Supplemental Data 3).

In order to evaluate for enrichment of putative TF binding site motifs, we used Homer analysis of pMRRs which identified motifs for TFs known to play prominent roles in melanoma, including SOX10[18,44,45] (p-value $= 1 \times 10^{-472}$) and ETS family factors[46], as well the multifunctional chromatin regulator CTCF (p-value $= 1 \times 10^{-6092}$, Supplemental Fig. 1a). However, pMRRs that encompassed statistically significant hotspots are only enriched in ETS motifs and are linked to a signature of UV damage, as previously observed[47] (Supplemental Fig. 1a). ETS factor motifs are not enriched in the mutant sequences, suggesting that most mutations break ETS transcription factor motifs (Supplemental Fig. 1a). We found an almost identical distribution of mutations around the canonical GGAA ETS motif within the significant hotspots identified in our pipeline as previously reported[47] (Fig. 1e).

To focus our efforts on a candidate(s) among the top scoring hotspots (i.e. those with scores higher than TERT, encompassing thirteen candidates), we evaluated gene expression for the gene most proximal to the hotspot between WT and mutant samples (Supplemental Fig. 2). We only considered variants that were present in at least three samples and observed statistically significant (t-test p-value < 0.05) changes in expression between WT and mutant variants in hotspots near CANX, CDC20, TERT, DPH3, PES1, RPS27, SLC30A6, RPL13A, and HNRNPUL1 (Supplemental Fig. 2). To refine our focus, we also looked for

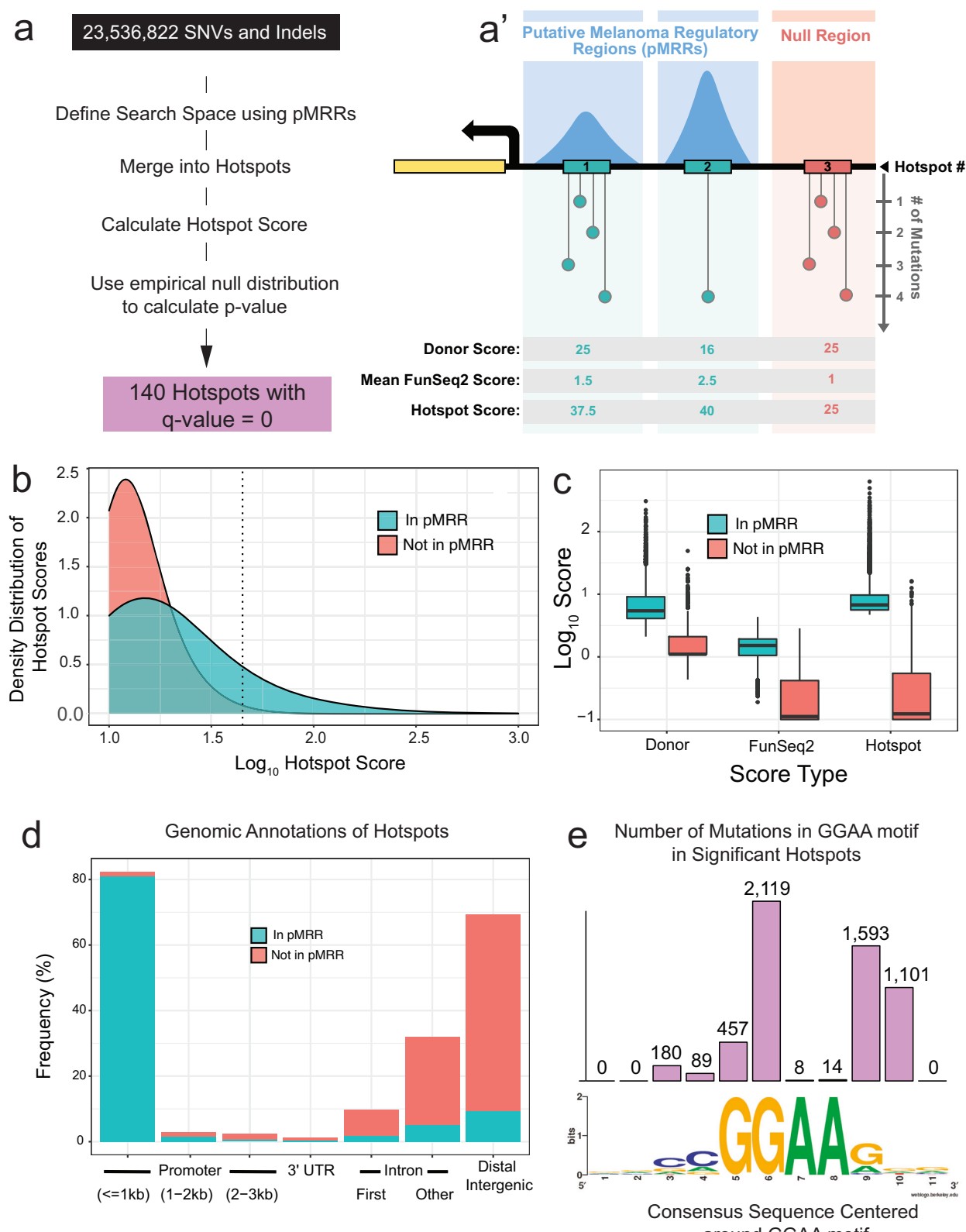

consistent changes in gene expression for the gene nearest the recurrent variants between different stages of melanomagenesis (Supplemental Fig. 1b). We used RNA-sequencing from five studies to calculate the fold change of the genes nearest to the hotspot between primary and metastatic tumors (The Cancer Genome Atlas, TCGA-SKCM[11] and ICGC-MELA[12]), nevi and melanoma (Kunz[48]), engineered melanocytes capable of forming

tumors and those incapable of forming tumors upon subcutaneous injection into immunocompromised mice (Hodis[5], see Methods for description of samples), and hPSC-derived melanoblasts with (KO melanoblasts) and without (WT melanoblasts) deletions in key tumor suppressors (Baggiolini[49], see Methods for description of samples). *CDC20* (gene associated with the 8th highest-scoring hotspot) is consistently upregulated in expression

**Fig. 1 A method to identify putative functional non-coding variants in human melanoma. a** Summary of pipeline to identify hotspots (**a**) with a generalized schematic of three theoretical hotspots (**a**). Blue boxes indicate regions within putative Melanoma Regulatory Regions (pMRRs), and red box indicates null regions (i.e. those outside predicted regulatory regions). Numbered rectangles represent hotspots. Dot plots represent the number of variants within a given position. Donor score is equal to the square of the number of donors divided by the number of mutated positions, and FunSeq2 score is a weighting factor with higher values indicating higher conservation within regulatory regions and/or TF binding site motif altering. **b** Kernel density estimate of hotspot scores in pMRRs (blue) and not in pMRRs/in null regions (red). Hotspots with $\log_{10}$ scores lower than 1 are not shown. Dashed line depicts hotspot scores with a $p$-value $= 1 \times 10^{-6}$, lower $p$-values are to the right. **c** Boxplots showing the $\log_{10}$-transformed Donor, FunSeq2, and Hotspot (Donor x FunSeq2) for the top 10,000 highest-scoring hotspots. **d** Bar chart demonstrating the frequency of genomic annotations for the top 10,000 null hotspots (red bars) and statistically significant hotspots (707 hotspots, FDR-adjusted $p$-value < 0.05, blue bars). **e** Bar chart of the total number of mutations in significant hotspots (707 hotspots) at each site within 4 bp of the core ETS motif, GGAA (top, represents 5,561 mutations out of a total 8514 mutations), and WebLogo of 11 bp WT sequence (bottom).

between melanoma and nevi (Kunz) and the KO and WT melanoblasts (Baggiolini, Supplemental Fig. 1b). We observe a small increase in metastatic tumors compared to primary tumors in the ICGC cohort and no change between primary and metastatic tumors in the TCGA. The only other $\log_2$ fold-change greater than 1 is seen in the ICGC cohort for *TERT* expression (increase in metastatic melanoma, Supplemental Fig. 1b). We did not observe any statistically significant associations with overall survival between tumors with and without hotspot mutations (Supplemental Fig. 3). Therefore, we stratified survival by expression and observed that low levels of *RPL18A* (3rd highest-scoring hotspot), *HNRPNUL1* (6th), and *CDC20* (8th) tumors have higher survival rates than tumors with high expression of these genes (Supplemental Fig. 1c). Taking expression changes between mutation status, differential gene expression of the nearest gene, and association with survival rates for those with melanoma into consideration, we specifically focus on characterizing the *CDC20* promoter in melanoma.

**Variants in the CDC20 promoter have different effects on its transcriptional regulatory activity.** The CDC20 promoter is mutated in 39 of 183 donors in the ICGC dataset, 38 of which are skin cutaneous melanomas (27.9% of cutaneous melanomas) and one acral melanoma sample (2.9% of acral melanomas). The most common single-nucleotide variants (SNVs) are at adjacent positions chr1:43,824,528 (G > A, hereinafter termed G528A, mutated in 10 donors) and chr1:43,824,529 (G > A, G529A, 16 donors) as well as a SNV at position chr1:43,824,525 (G > A, G525A, 4 donors) and a multi-nucleotide variant (MNV) at positions chr1:43,824,528-43,824,529 (GG > AA, GG528AA, 4 donors) and are located within an ETS motif (Fig. 2a). While at adjacent positions, G528A and G529A have different FunSeq2 scores (second number) and Genomic Evolution Rate Profiling (GERP) scores (third number) suggesting different degrees of purifying selection[50]. The singular acral melanoma with a CDC20 promoter hotspot mutation contained the G529A variant. G525A is located within the core ETS motif, at the position that is most often mutated when taking all variants within all statistically significant hotspots into consideration (Fig. 1e) but is not the most recurrent variant in the CDC20 promoter hotspot, occurring only in 4/39 donors. Like G528A, G525A has both a high FunSeq2 score and a high GERP score.

Samples from the ICGC-MELA cohort that were RNA-sequenced containing the GG528AA CDC20 promoter mutation had a significant reduction of overall *CDC20* expression compared to WT ($p$-value $= 0.05$, Student's $t$-test) while samples with the G529A mutation were not significantly different ($p$-value $= 0.14$, Student's $t$-test, Supplemental Fig. 2). We note that due to low sample size there were not enough samples containing the G525A and G528A mutations to assess variant-specific changes in CDC20 expression. As previously observed in the TCGA-SKCM cohort[11], only samples containing the C228T TERT promoter mutations

were statistically significantly associated with increased expression (Supplemental Fig. 2).

CDC20 promoter hotspots are not more likely to co-occur with pathogenic *BRAF* mutations than *NRAS* ($p$-value $= 0.85$, Fisher's Exact Test, Supplemental Fig. 4a) nor were they more likely to co-occur with pathogen mutations compared to wild-type for both BRAF and NRAS (BRAF $p$-value $= 0.28$, NRAS $p$-value $= 0.81$, one-sided Fisher's Exact Test, Supplemental Figs. 4b, c). However, the presence of a CDC20 promoter hotspot mutation or a TERT promoter hotspot mutation was significantly associated with UV signature ($p$-value $= 2.244 \times 10^{-7}$ for CDC20 and $p$-value $= 5.039 \times 10^{-14}$ for TERT promoter, Wilcoxon Test, Supplemental Figs. 4d, e). Although not conclusively caused by UV irradiation, this is in line with statistically significant hotspots frequently occurring at actively bound ETS motifs as they are particularly susceptible to UV irradiation[47,51].

Overlaying chromatin-related assessments of the locus, the CDC20 promoter is accessible in 4/7 datasets that assay genome-wide chromatin accessibility (Supplemental Data 1). BRG1, CTCF, and TFAP2A are among the chromatin/transcription factors that have binding activity at the CDC20 promoter, as detected by ChIP-seq. ETV1, the only ETS factor with ChIP-seq data in our collation of melanoma-specific functional datasets, did not have binding activity at the CDC20 promoter in the 2 cell lines assayed (A375 and COLO-800, Supplemental Data 1).

To understand how the variants affect the regulatory activity of the CDC20 promoter, we performed luciferase assays using a 150 bp sequence encoding the CDC20 promoter hotspot in a promoter-less luciferase vector. We assayed the three most prevalent variants, G528A, G529A, and GG528AA, all located near the canonical ETS motif, and two variants not within the motif that were detected in one donor, C520T and C527T. We investigated the effect of these variants on luciferase activity relative to the wild-type promoter in seven melanoma cell lines which harbor previously identified distinct transcriptional identities that range from melanocytic (i.e. more proliferative) to undifferentiated (i.e. more migratory, Fig. 2, Supplemental Fig. 5a, Supplemental Data 4, Supplemental Data 5)[20]. We also performed luciferase assays in primary melanocytes isolated from neonatal foreskin and HEK 293FT, a human embryonic kidney cell line. All melanoma cell lines harbor the BRAF[V600E] mutation except for SK-MEL-2 which contains the NRAS[Q61R] mutation.

Most notably, G525A, located within the canonical ETS motif, and G528A, located adjacent to the motif, consistently and significantly reduced luciferase activity in all melanoma cell lines except for the G528A mutation in SK-MEL-2 cells which showed an increase (Fig. 2b, Supplemental Data 4). Interestingly, G529A did not always have similar effects on reporter activity as G528A, despite being only one nucleotide further from the ETS motif than G528A. For example, while we observed similar reductions in luciferase activity between G528A and G529A in UACC-62 and RPMI-7951, we observed a slight increase in G529A in LOX-

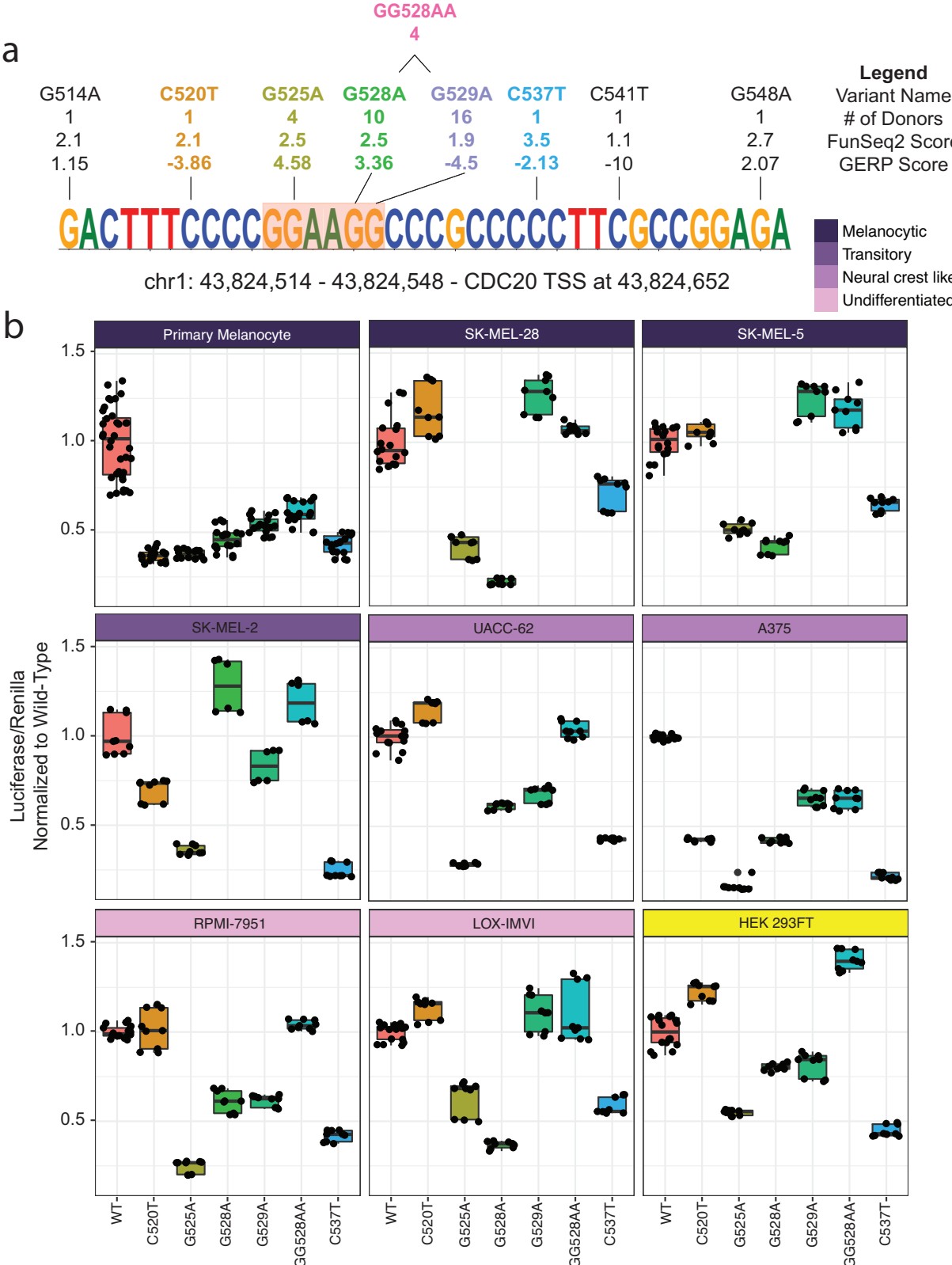

**Fig. 2 Functional analysis of recurrent CDC20 promoter variants. a** The CDC20 promoter hotspot. All variants within the hotspot are denoted by name, # of donors with given mutation, FunSeq2 score, and GERP score. Co-occurring GG528AA double mutant is depicted above. Variants with colored text were validated by luciferase assay. **b** Altered CDC20 promoter activity for variants as assayed by luciferase reporter assays in nine cell lines, ordered by transcriptional state classification from melanocytic to undifferentiated. HEK 293FT are immortalized human embryonic kidney cell lines and are therefore not classified. Boxplots depict normalized (to WT) luciferase assay results in these nine different cell lines.

IMVI, SK-MEL-5, and SK-MEL-28 (Fig. 2b, Supplemental Data 4). Moreover, when we consider the effects of the multi-nucleotide variant GG528AA, we observe an effect that is less deleterious than G528A, only leading to a significant reduction of luciferase activity in A375 and primary melanocytes, and even a significant increase in the non-melanoma cell line HEK 293FT, as previously observed[10]. Finally, we did not observe changes in activity that associated with the transcriptional identity of the cell line, suggesting the effect of the variants is independent of the cell identities used here. For example, LOX-IMVI (i.e. undifferentiated) and SK-MEL-5 (i.e. melanocytic) had similar luciferase assay results despite stark differences in transcriptional states.

We evaluated the expression levels of ETS transcription factor family members in melanoma but did not observe a pattern that implicates any one transcription factor to the observed effects of the luciferase assay (Supplemental Fig. 5b). Therefore, to interrogate the binding activity at the CDC20 promoter hotspot, we leveraged existing ENCODE transcription factor ChIP-seq data from multiple cell types. While these data did not include melanoma cell lines or samples due to inadequate representation of this sample type in ENCODE (Supplemental Data 6), we assume that transcriptional regulation will share core mechanisms across cell types due to the ubiquitous and essential function of CDC20. We observe binding of several ETS factors, predominantly ELF1 and GABPA which are known to be expressed in melanoma[52], as well as other factors known to play important roles in melanoma, YY1, JUN, and MYC (Supplemental Fig. 6a)[53–56]. siRNA-mediated knockdown of ELF1 but not GABPA in A375 melanoma cells led to a significant reduction of CDC20 (Supplemental Fig. 6b).

We then generated position weight matrices (PWM) using a 10 bp sequence encapsulating the CDC20 promoter hotspot ETS motif (Fig. 2a, orange highlighted box, motif includes the flanking 2 bp) in the presence of the G525A, G528A, G529A, or GG528AA variants. We used Homer to scan these PWMs against our collated pMRRs. We expected, as observed in our luciferase assays, sequences that were similar to the G525A mutation would have the most dramatic reduction in binding activity while G529A and GG528AA would not be as affected. Indeed, for ELF1, we saw that sequences that resembled the CDC20 promoter mutations reduced binding activity with G528A, G525A, and GG528AA leading to the largest reductions (Supplemental Fig. 6c). In the case of GG528AA, we did not expect such a drastic reduction as the GG528AA mutations were less deleterious than the G528A, and in some cases the G529A mutations, in the luciferase reporter assays. Interestingly, when we looked at GABPA binding activity, we saw that only the G525A CDC20 promoter mutation had an observable reduction in binding activity (Supplemental Fig. 6d). Taken together, this suggests a putative regulatory role for ELF1 on CDC20 expression.

**Distinct transcriptional programs emerge in nevi and melanoma in a CDC20 dosage-associated manner.** CDC20 promoter variants lead to a decrease in reporter activity; however, tumors with relatively higher levels of CDC20 have worse overall survival (Supplemental Fig. 1) and CDC20 has been shown to be essential for migration in melanoma mouse models[57]. Therefore, we hypothesize the existence of both a CDC20-low and CDC20-high phenotype that may support specific transcriptional programs associated with key melanoma phenotypes, as is known with MITF[15,58,59]. To begin probing for the potential existence of a CDC20-high and CDC20-low state, we utilized three cohorts of bulk RNA-sequencing on melanoma tumors: (1) Kunz et al. (2018) cohort of 23 nevi and 57 primary melanomas, (2) TCGA-SKCM, and (3) ICGC-MELA. We performed gene set enrichment analysis[60] (GSEA) on samples with relatively high and relatively

low levels of CDC20 using published gene signatures that stratify melanomas into two major subtypes: (1) highly proliferative/weakly metastatic and (2) weakly proliferative/highly metastatic (Supplemental Data 7)[61,62].

The gene expression program of CDC20-low samples is more enriched for genes in the neural crest-like and proliferative gene sets based on relatively high expression of genes like SOX10, an important regulator of the melanocytic and neural crest lineage (Fig. 3a, Supplemental Fig. 7). Up-regulation of SOX10 is believed to, at least in part, contribute to the re-emergence of the neural crest features observed during melanoma initiation and progression[18,19,21,44]. The gene expression program in CDC20-high samples is enriched for genes in the invasive and TGFβ-like gene sets which contains genes like AXL and AMIGO2 that have been positively associated with invasion and metastasis (Fig. 3b, Supplemental Fig. 7, Supplemental Data 7)[15,22,63].

Additionally, we leveraged publicly available single-cell RNA-sequencing datasets of melanoma cell lines and tumors with sufficient coverage of CDC20 for further observations of a proliferative CDC20-low transcriptional state and an invasive CDC20-high transcriptional state. We observed a statistically significant difference ($p$-value $< 2.2 \times 10^{-16}$) in the expression of CDC20 between the cells classified as relatively more melanocytic (lower levels of CDC20) compared to the clusters of cells classified as more mesenchymal-like[13,59] (higher levels of CDC20, Fig. 3c, Supplemental Data 8). This trend was also observed in a secondary dataset that classified cells using a separate but related MITF signature[15,64] (Fig. 3d-e, Supplemental Data 8).

Taken together, these results set the precedent for a population of CDC20-low and CDC20-high cells that may drive transcriptional programs more favorable at specific stages of melanoma. CDC20-low populations appear to adapt a melanocytic/proliferative gene signature, commonly seen in primary stages of melanoma, whilst CDC20-high populations may benefit migration/metastasis.

**Genome-engineered CDC20 promoter mutants have altered phenotypes and transcriptional profiles.** Thus far, we have identified variants prevalent in the CDC20 promoter in melanoma tumors that by luciferase reporter assay reduce transcriptional activity and see distinct transcriptional profiles in naturally occurring human melanoma tumors and nevi associated with high and low levels of CDC20. To determine the effect of CDC20 promoter mutations on key cancer phenotypes and gene expression programs, we generated two CRISPR/Cas9-engineered A375 melanoma cell lines termed A3 and A10 (Fig. 4a). The A3 line contains an indel on both alleles, both of which have the G528 and G529 nucleotides deleted. One allele retains the core GGAA motif while the other does not. The A10 line contains a larger deletion that completely removes the G525, G528, and G529 mutations, as well as the core ETS motif in both alleles (Fig. 4a).

Both indels decrease CDC20 expression by 2.0-fold on average as detected by RNA-sequencing (FDR-adjusted $p$-value $= 1.8 \times 10^{-40}$, Fig. 4b, Supplemental Data 9). The A3 strain has slightly lower CDC20 expression than A10 despite having a smaller deletion and the retention of one core ETS motif (Fig. 4b). Principal component analysis shows a separation along PC1 between the WT parental A375 line (high CDC20), the WT Cas9 control A375 line (high CDC20), and the mutant A3 and A10 line (low CDC20, Fig. 4c).

We identified 3223 differentially expressed genes using bulk RNA-seq between both WT and both mutant cell lines (Supplemental Data 9). There were 1,955 and 999 differentially expressed genes between A10 and both WT strains and 999 between A3 and WT (Supplemental Data 9). However, by hypergeometric test, the number of overlapping genes is significant ($p$-value $< 2.2 \times 10^{-16}$).

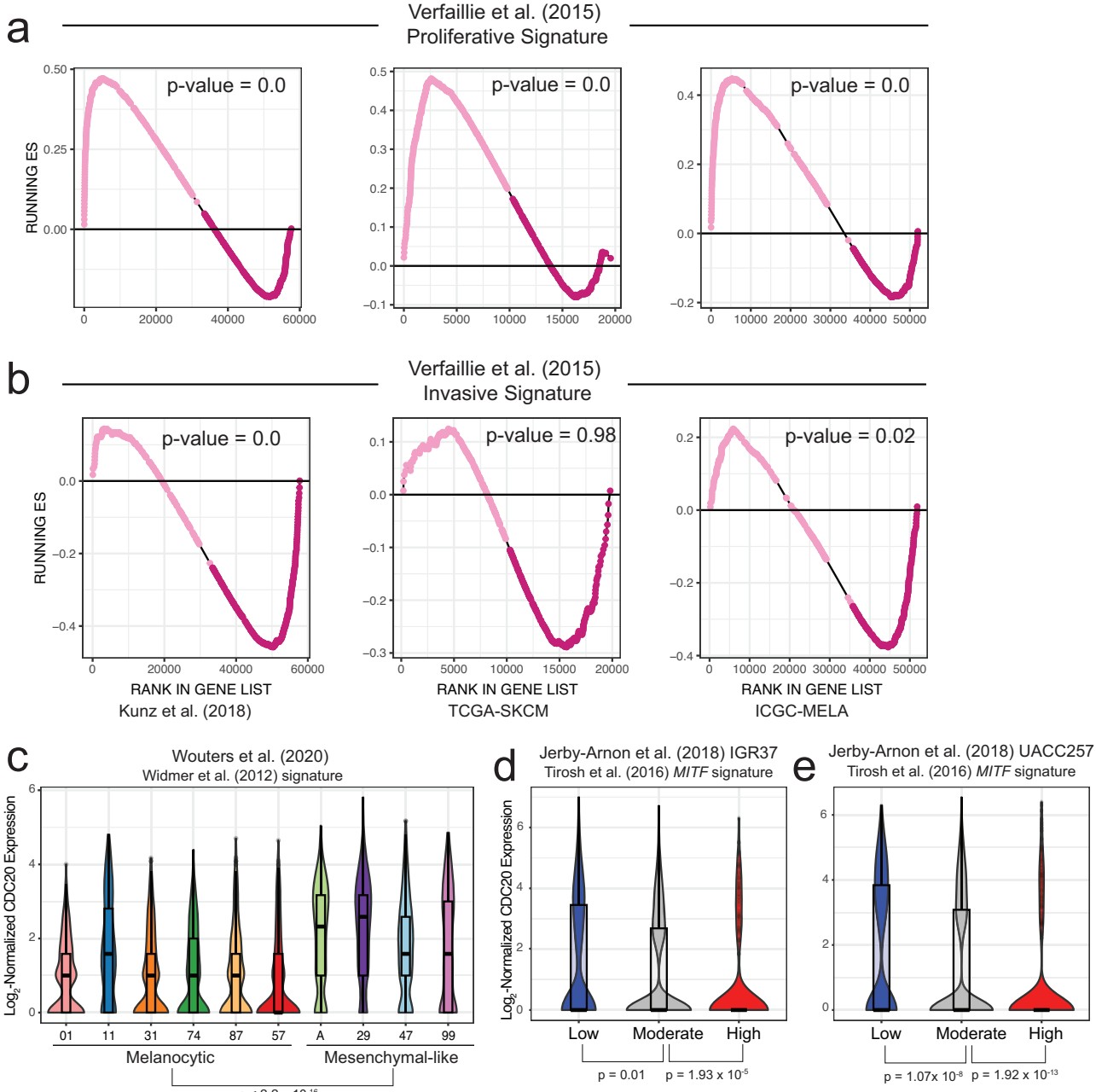

**Fig. 3 CDC20-low and CDC20-high transcriptional states correlate with distinct melanoma signatures. a, b** Results from gene set enrichment analysis for the Verfaillie proliferative (**a**) and invasive (**b**) melanoma gene signature. Light pink dots represent genes enriched in *CDC20*-low samples (samples with *CDC20* expression at or below the 25th quantile) while dark pink dots represent genes enriched in *CDC20*-high samples (samples with *CDC20* expression at or above the 75th quantile). Samples are from three independent RNA-sequencing cohorts: Kunz, TCGA-SKCM, and ICGC-MELA. Nominal *p*-values are reported. **c** Normalized *CDC20* gene expression obtained from scRNA-sequencing of 10 melanoma samples ordered by decreasing melanocytic signature (Widmer et al. 2012) as calculated by AUCell in the corresponding publication[13,59]. A Wilcoxon rank-sum test was performed on *CDC20* expression between all samples classified as melanocytic and those classified as mesenchymal. **d, e** Normalized *CDC20* gene expression from melanoma cell lines (**d**) IGR37 and (**e**) UACC257. Each cell was given an MITF signature score based on the MITF signature from Tirosh et al. (2016), with high meaning cells expressing relatively higher levels of an MITF transcriptional program. Wilcoxon rank-sum tests were performed between low and moderate and moderate to high[15,64].

Using the differentially upregulated genes between each mutant strain and both together results revealed an enrichment of gene ontology (GO) terms that were mostly similar across all comparisons (Fig. 4d). Notably, the A3 CDC20 promoter indel line lacked genes involved in Wnt signaling.

We performed GSEA on the A375 WT and CDC20 promoter indel lines A3 and A10 using the same gene sets as before

(Figs. 3a and 2b and Supplemental Fig. 7). There is a significant enrichment of genes upregulated in CDC20 promoter indel cells vs WT for genes in the Verfaillie proliferative and Hoek neural crest-like gene signature[61,62] (Fig. 4e). For WT lines, we noted significant enrichment in the Verfaillie invasive gene signature but an unsignificant enrichment of genes in the Hoek TGFβ-like gene signature (Supplemental Data 7).

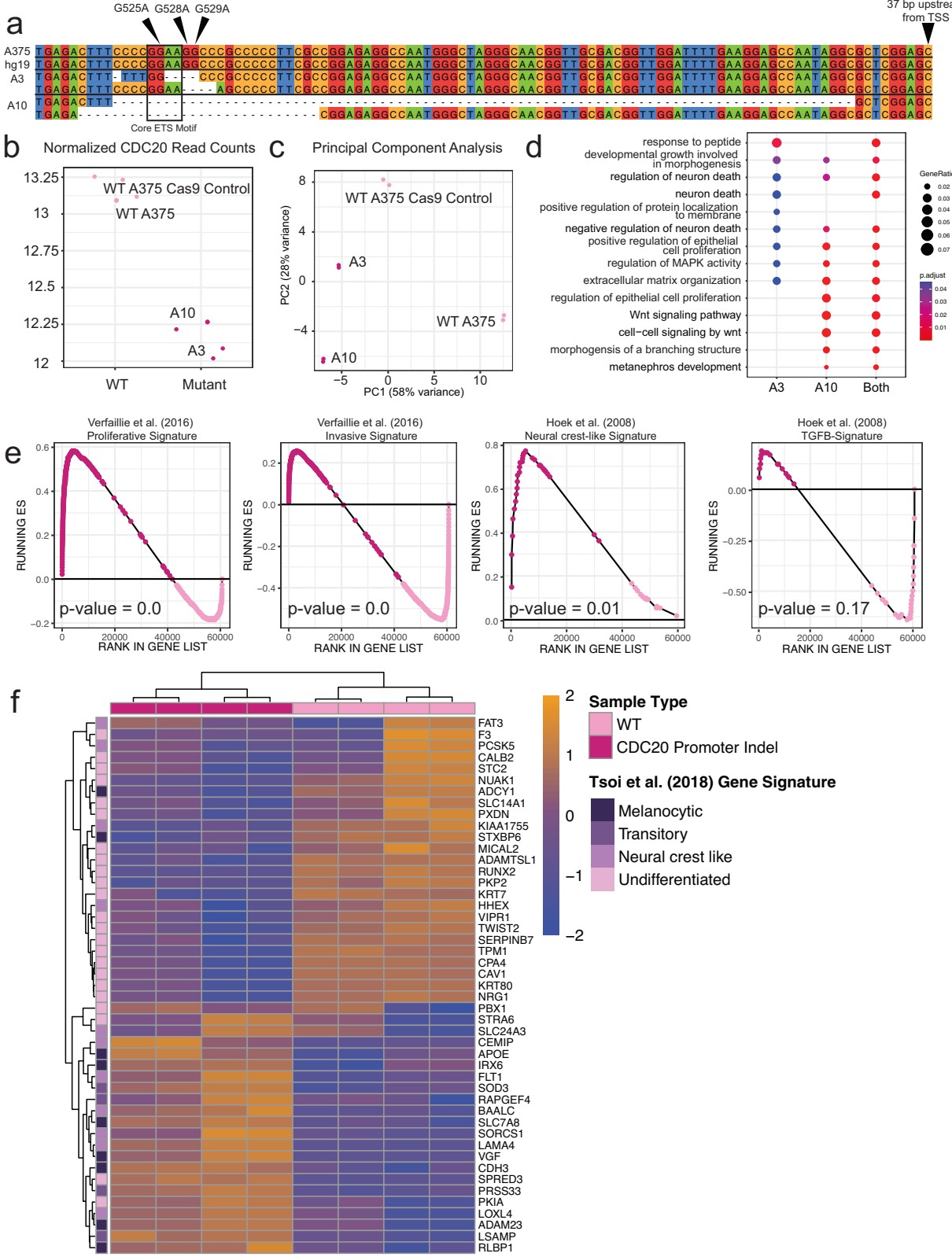

We used a third gene signature that stratifies melanoma by four subtypes (i.e. melanocytic, transitory, neural crest-like, and undifferentiated) to cluster the differentially expressed genes between WT and CDC20 promoter indel cell lines[20]. As expected, based on the GSEA results and the known transcriptional profile of the A375 melanoma cell line, we observed that WT cell lines have relatively high expression of genes in the undifferentiated subtype

(Fig. 4f). On the other hand, the mutant cell lines, which have relatively lower expression of CDC20 due to the promoter variants introduced by CRISPR/Cas9, have relatively higher expression of genes associated with the melanocytic and transitory subtypes. These results support the notion that relatively lower expression of CDC20 promotes a phenotypic state that is more proliferative, less invasive, and contains relatively high expression levels of important

**Fig. 4 Engineered indels at the recurrently mutated CDC20 promoter locus leads to decreased *CDC20* expression and changes in transcriptional state.**
**a** Sequence alignment of the CDC20 promoter between hg19, WT A375, A3, and A10. Arrows denoting positions of G525A, G528A, and G529A. The ETS core motif is boxed. The last nucleotide of the sequence is 37 bp upstream of the TSS of *CDC20*. Nucleotides are color-coded and dashes indicate deletions.
**b** Plot depicting log$_2$ transformed DESeq2-normalized read counts of *CDC20* in WT A375 and CDC20 promoter indel strains, A3 and A10, with decreased *CDC20* expression. Each point represents *CDC20* expression in one sample. **c** Principal component analysis of read counts normalized by regularized log transformation using the top 500 most variable genes. The horizontal axis, PC1, explains 58% of the variance associated across all samples and separates out WT from CDC20 promoter indel cell lines. The vertical axis, PC2, explains 28% of the variance and separated A3 from A10. **d** Gene ontology analysis of the 999 differentially expressed genes (DEG) between A3 and WT, 1,995 DEG between A10 and WT, and 3,223 DEG between both CDC20 promoter indel cell lines and WT. The size of the dots represents higher gene ratios of the number of genes within each gene ontology term. The color of the dot represents FDR-adjusted *p*-values with redder dots indicating lower *p*-values. **e** Gene set enrichment analysis of the Hoek and Verfaillie gene signatures with nominal *p*-values reported on each plot. Dark purple dots represent genes enriched in the CDC20 promoter indel cell lines while light purple dots represent genes enriched in the WT A375 melanoma cell lines. **f** Heatmap depicting z-score normalized expression patterns of differentially expressed genes within one of the four Tsoi et al. (2018) melanoma transcriptional subgroups. Samples and genes are hierarchically clustered with orange and blue indicating relatively higher or lower expression, respectively, of genes across samples.

neural crest and melanocyte lineage specifiers such as *SOX10*. As *CDC20* levels increase, a transcriptional state that is more associated with invasion and metastasis is observed, including relatively higher expression of *AXL*, a gene associated with metastasis in melanoma and other cancers[15,65].

To extend these hypotheses to other melanoma cell lines, we knocked down *CDC20* by siRNA and performed quantitative PCR (qPCR) on *SOX10* on melanoma cell lines spanning multiple subtypes. In all melanoma cell lines assayed, siRNA-mediated knockdown of *CDC20* significantly upregulated *SOX10* (Fig. 5a, Supplemental Fig. 8, Supplemental Data 10). We also assessed how *CDC20* knock-down would affect *AXL* expression in melanoma cell lines with relatively high levels of *AXL* (i.e. those classified as undifferentiated). Interestingly, in these cells, we observed a significant decrease in *AXL* (Fig. 5a). This lends support to our hypothesis that low level of CDC20 supports a more proliferative and less invasive state.

CDC20 mutations have previously been shown to cause aneuploidy[66–68]. We therefore analyzed all melanoma cell lines in the Cancer Dependency Map cohort (DepMap) and melanoma tumors in TCGA to examine whether *CDC20* expression correlates with aneuploidy. We did not observe a statistically significant association between *CDC20* expression and aneuploidy in either cohort (Supplemental Fig. 9a, b). Additionally, we checked the A375 mutant lines for increased aneuploidy but did not observe any in a karyotyping analysis (Supplemental Fig. 9C).

To see whether our A375 promoter indel lines have altered migration capabilities as suggested by the results of GSEA and the literature[57], we performed a Boyden chamber assay containing collagen and a scratch assay and observed decreased migration capabilities in both experiments suggesting that, at least in this context, reduced levels of *CDC20* affect migration more so than viability (Fig. 5b, c).

Because of the essential role of CDC20 in the cell cycle and the more proliferative transcriptional signature of *CDC20*-low samples, we assayed changes in growth rates in the presence of media containing serum, media containing serum and DMSO, and media containing serum and 30 nM dabrafenib (MAPKi) daily over the span of 6 days (Supplemental Fig. 9d). Minimal changes in growth rates were observed with slightly higher numbers of A3 cells in culture by day 6 and initially slightly lower cell numbers for A10 than A375 WT and A3.

Finally, we assessed whether the CDC20 promoter indel lines have altered growth in an in vivo context using xenografts of the CDC20 indel and WT lines in the flanks of immunodeficient nude mice. We generated xenografts for WT A375 and A3 and A10 CDC20 promoter indel melanoma cell lines (5 mice per cell line, 2 flank tumors per mouse, 3 cell lines for a total of 30 total tumors) and measured tumor volume over the course of 6 weeks with

unblinding only after completion of all measurements and analysis (Fig. 5d, Supplemental Data 11). We observed a significant increase in the tumor volume of A3 (average tumor volume at 5th timepoint = 1316 mm$^3$, standard error mean = 66.1 m$^3$, q-value = 0.002) and a slightly larger but unsignificant increase in the tumor volume of A10 (576 mm$^3$, 48.9 mm$^3$, q-value = 0.30) as compared to WT A375 parental line (386 mm$^3$, 36.8 mm$^3$) by 6 weeks of xenograft growth (Fig. 5d). In conclusion, we demonstrate that non-coding indels in the promoter of CDC20 reduce *CDC20* expression, up-regulate genes associated with a more proliferative signature, and increase xenograft growth, supporting a putative cancer-driving role for the non-coding single nucleotide variants identified in 27.9% of skin cutaneous melanomas.

## Discussion

Using the largest available cohort of melanoma whole-genome sequencing data and several dozen melanoma-specific functional genomics datasets, we have identified hundreds of mutational hotspots containing putatively functional non-coding somatic variants. Under the assumption that variants outside of pMRRs are not, or are less likely to be, functional, we generated an empirical null distribution with which to calculate significance. Several of our top-scoring hotspots have previously been identified as being in regions of the genome that are recurrently mutated due to presence of a motif sensitive to UV mutagenesis at actively bound promoters[32,47,51]. Although none of the hotspots were significantly associated with overall survival using TCGA data, which while an extremely valuable and often informative resource, overrepresents nodal metastases relative to primary and other distant metastases types[11], we did identify promising changes in the relative expression of hotspot-associated genes between WT and mutated samples, despite being relatively under-powered due limited RNA-sequencing. Shifting our analysis instead towards gene expression, we observed expression changes of the genes nearest to the hotspots between early and late tumors, as well as in overall survival of tumors with relatively higher or lower gene expression. Although our analysis identified other promising candidates, including the presence of several ribosomal genes, one of which has been previously validated[9], we chose to focus on characterizing variants in the promoter of CDC20, whose weighted rate of recurrence and predicted functional significance were greater than that of the well-studied *TERT* promoter variants and began to investigate how these variants alter melanoma behavior.

CDC20 is a highly conserved and essential regulator of the cell cycle. Deletion of CDC20 leads to arrest at metaphase in 2-day old embryos in mice[69]. It is a catalytic co-activator of the Anaphase Promoting Complex/Cyclosome (APC/C) which is a large

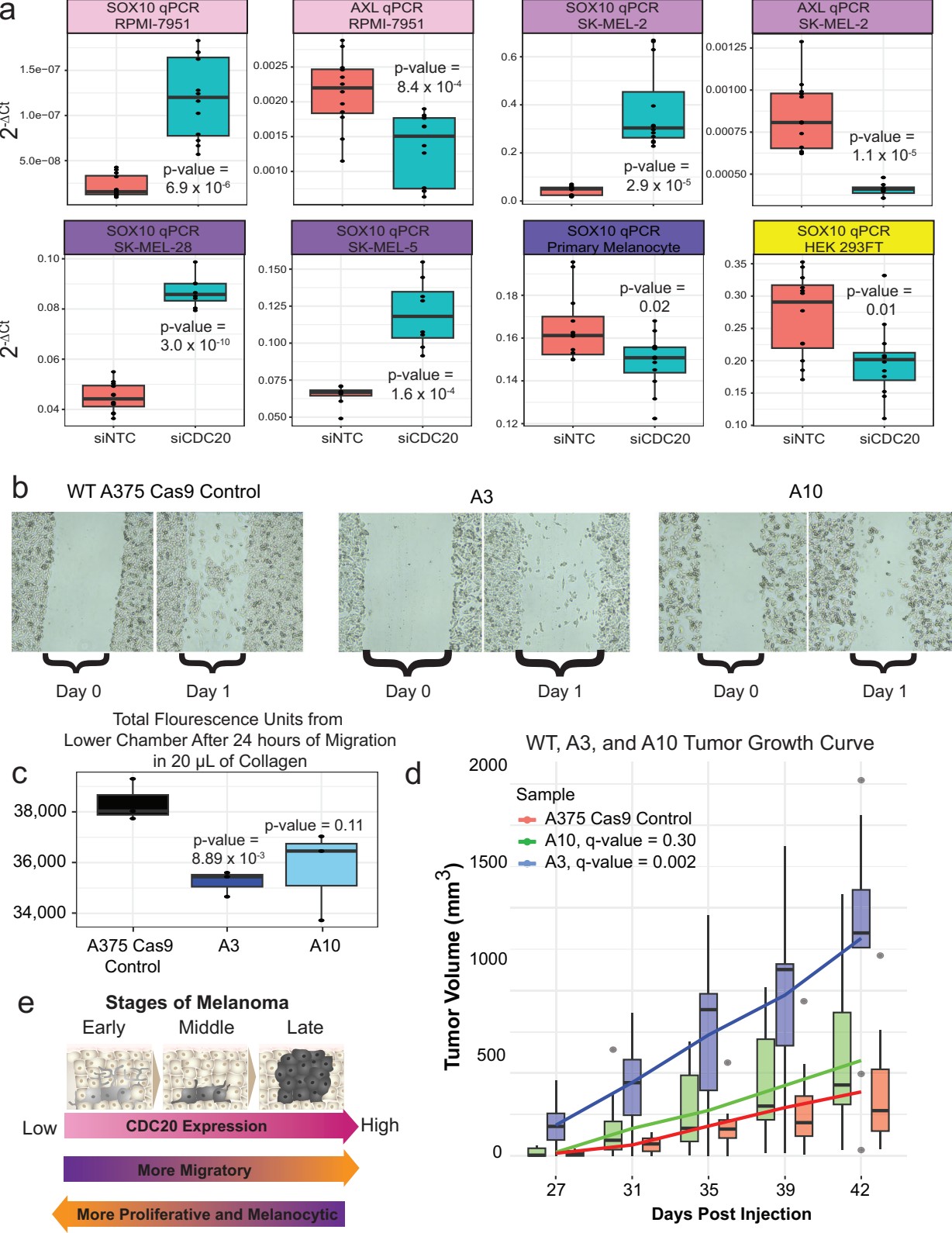

multi-subunit E3 ubiquitin ligase. CDC20 contains many degrons (i.e. protein 'motifs' that recognize and target substrates for degradation) for proteins including Cyclin B1 and Securin[70].

Degradation of Cyclin B leads to the inactivation of mitotic cyclin-dependent kinases and degradation of securin leads to activation of separase, causing separation of sister chromatids. The combined effect of the two mark the beginning of anaphase and end of mitosis[71]. The spindle assembly complex (SAC) binds to CDC20 and MAD2 (forming the mitotic checkpoint complex, MCC) at the kinetochore to inhibit APC-CDC20 until all sister chromatids are attached to the kinetochore[72]. Cyclin-dependent kinases also phosphorylate CDC20 which reduces the binding efficiency with APC/C, therefore preventing APC/C-CDC20 formation and G2-to-M transition[73]. This is an essential function

**Fig. 5 CDC20 knockdown promotes a proliferative or neural crest transcriptional state. a** Knockdown of *CDC20* by siRNA leads to significant down-regulation of *AXL* and up-regulation of *SOX10* in undifferentiated cell lines RPMI-7951 and SK-MEL-2, up-regulation of *SOX10* in melanocytic cell lines SK-MEL-5 and SK-MEL-28, but no to minimal change in primary melanocytes and HEK 293FT. Each boxplot is the culmination of 12 technical replicates transfected with a non-targeting siRNA (siNTC) or siCDC20. Ct scores are normalized to *GAPDH*. **b** Scratch assays at 0 hours post scratch and 24 hours post-scratch for WT, A3, and A10. **c** Total fluorescence obtained from the lower chamber of a Boyden chamber loaded with 20 μL of collagen. Three technical replicates were performed per cell line. **d** Tumor volume (average of 10 tumors per line with error bars ± SEM) of WT (A375 Cas9 control), A3, and A10 between 27 and 42 days post subcutaneous flank injection into immunocompromised nude mice. **e** Proposed model. As *CDC20* expression increases, a more invasive and migratory transcriptional landscape is observed, suggesting functional significance for CDC20 in the role of metastasis. At early stages of the melanoma lifespan, low *CDC20* expression promoters a proliferative state that fuels primary melanoma growth.

as the inability to inhibit APC-CDC20 leads to rapid tumor-igenesis in mice[74].

CDC20 has also been implicated in functions outside of the SAC and MCC, including the mediation of chromatin loop formation through ubiquitylation of hnRNPU[75], attenuation of cardiac hypertrophy[76], maintenance of the primary cilia[77], regulation of dendrite morphogenesis in neurons[78,79], and most relevant to our study, maintaining stemness in human primary keratinocytes and human embryonic stem cells[80,81]. The Bio-GRID database (thebiogrid.org) has 231 unique interactors cataloged for CDC20, including histone remodelers HDAC1/2, chromatin remodeler CTCF, and neural crest transcription factor and oncogene MYC. Thus, understanding how a reduction in *CDC20* appears to favor a melanocytic/proliferative state in melanoma will require substantial further investigation and appears warranted based on our findings for its role in melanoma given that it may alter the ubiquitylation kinetics of the APC/CDC20 complex and many other targets.

Specific CDC20 promoter variants tested decreased reporter activity across most cell lines in this study. Four variants were within 2 bp of a core ETS motif but did not all affect reporter activity to similar extents. G525A and G528A strongly reduced reporter activity in most cell lines assayed, while G529A and GG528AA reduced reporter activity in a subset of cell lines, suggesting a cell-specific response to these variants. We favor that these differences in transcriptional responses are influenced by specific combinations of transcription factor activities (that is variations in protein levels, modifications affecting abilities to enhance or repress gene expression, and interactions with and availability of necessary cofactors), as well as potentially more broadly defined DNA and histone modifications, that are likely specific to different melanoma cell lines and to distinct stages of the melanocyte to melanoma transition which, even when assayed in in vitro models, are likely transient and difficult to directly probe. Interestingly, the CDC20 promoter has not been reported to be methylated in previous work[82–86]. To begin investigating the binding activity at the CDC20 promoter hotspot, we used ENCODE ChIP-seq data from several non-melanoma cell lines and identified potential binding of multiple transcription factors, including ETS family members, which were predicted to have altered binding in naturally occurring motifs that matched the mutated ETS motif at the CDC20 promoter when mutated (Supplemental Fig. 6). In particular, we observed binding activity of ELF1 at the CDC20 promoter and a reduction in *CDC20* upon *ELF1* knockdown by siRNA in a melanoma cell line. While *ELF1* is known to be expressed in melanoma[52], future work will similarly be required to understand the exact mechanism by which CDC20 promoter mutations induce differential transcriptional regulation, likely by specifically altered transcription factor binding activity in distinct, dynamic melanoma states. This study lends strong support for continued endeavors into ELF1 regulation of CDC20.

We identified both in our genome engineered cells and in three other naturally occurring human melanoma cohorts that samples

with low *CDC20* express transcriptional programs associated with a neural crest-like or proliferative state, characterized by relatively higher expression of genes like *SOX10*, which was then validated by siRNA-mediated knockdown of *CDC20* in multiple melanoma cell lines (Fig. 5a, Supplemental Data 9, Supplemental Data 10). However, we did not observe this up-regulation in primary melanocytes nor the non-melanoma cell line HEK 293FT, despite functional validation of the CDC20 promoter hotspot mutations in luciferase reporter assays in these cell lines. We speculate that both early oncogene activation and specific lineage identity is required for the effect of the CDC20 promoter hotspots, through downregulation of *CDC20*, on promoting a transcriptional program that, based on previous reports, is important for establishment of primary melanomas[18,19,21,44]. In support of this hypothesis, we observed a statistically significant increase in tumor growth of the A3 CDC20 promoter indel cell line relative to the parental A375 cell line when subcutaneously xenografted in the flanks of immunocompromised *nude* mice (Fig. 5d). Interestingly, the growth rates were markedly faster for A3 compared to wild-type in the in vivo study than as assessed by the in vitro cell viability assay, suggesting an important interaction between the tumor microenvironment and CDC20 function. Indeed, one of the GO terms enriched in the mutant cell lines implicate gene regulatory networks that function to alter the extracellular matrix (Fig. 4d). Meanwhile, as *CDC20* levels increase, we observe a change in the transcriptional program that is associated with the invasive subtype, including an increase in the expression of *AXL*[87] (Supplemental Data 9). Both in our CDC20 promoter indel lines and in two other melanoma cell lines that are characterized by relatively high expression of *AXL*, we observed a significant reduction of *AXL* upon *CDC20* knockdown by siRNA (Fig. 5a). Therefore, we propose cells with relatively high expression of *CDC20* may gain migration capabilities. In support of this, we observed loss of migratory capabilities in A3 and A10, the CDC20 promoter indel cell lines with lowered *CDC20* levels (Fig. 5c). As has been recently shown, transcriptional heterogeneity in primary melanomas may consist of cell types capable of driving primary tumor growth but not metastasis and vice versa[21]. With this ever-increasing recognition of tumor heterogeneity, especially in metastases[87,88], larger sample catalogs will be required to make claims about selective advantages/disadvantages of specific variants. Although we did not detect metastases in the mice injected with the wild-type A375 melanoma cell line, further study is warranted to understand the metastatic potential of *CDC20*-high versus *CDC20*-low expressing melanoma cell lines.

The ever-expanding genomic data available for melanoma has been crucial in advancing our understanding of melanoma biology[11,12], but most of the largest datasets with publicly available clinical outcomes data (i.e. TCGA) overrepresent metastatic lesions, and even then a subset of metastatic lesion types (i.e. lymph node metastases in TCGA). Thus, while *CDC20* has been implicated as a cancer-driving gene with higher levels often associated with melanoma metastases and poorer survival, we posit that specific levels of *CDC20* expression may be crucial to

supporting or allowing passage of melanocytes through malignant transformation (*CDC20* low) to locally invasive cancer and then on to metastatic disease (*CDC20* high, Fig. 5e). As in the case of MITF, a rheostat model of *CDC20* may exist, whereby higher levels of *CDC20* drives metastasis and lower levels support a phenotype likely beneficial in earlier tumors[58].

## Methods
### Calculating hotspot scores
*Step 1: Merge mutations into hotpots.* Mutation calls for SNVs and indels from the MELA-AU cohort were downloaded from dcc.icgc.org after receiving DACO approval[12]. Using a 25 bp window, we merged mutation calls using bedtools intersect[89] into hotspots based on the premise that highly recurrent variants may be under positive selection at some point during the melanoma life cycle (e.g. favor melanoma growth) and that a transcription factor binding site(s) (TFBSs) may be disrupted/created by modifying any of multiple nucleotides in this window.

*Step 2: Filter hotspots not in putative enhancers/promoters.* We downloaded processed peak calls from ChIP-seq (e.g. H3K27Ac, H3K4me3, CTCF) and ATAC-Seq (revealing accessible chromatin domains) data from 69 melanoma datasets to enrich for putative Melanoma Regulatory Regions (pMRRs) which we reasoned are more likely to bind transcription/chromatin factors (Supplemental Data 1). These are indicated by the blue peaks in the example Fig. 1a. We excluded exons and those regions (e.g. highly repetitive) from Encode excluded regions list[90].

*Step 3: Calculate donor score.* The donor score for a given hotspot is represented as $D^2/G$, where $D$ is the number of samples (donors) with the specific variant and $G$ is the number of nucleotide locations with variants in the hotspot. For example, in Fig. 1a, the purple hotspot shows $D = 3 + 1 + 2 + 4 = 10$ mutations, at $G = 4$ different locations, for Donor Score of $10^2/4 = 25$.

*Step 4: Weight variants using FunSeq2 score.* Each mutation is weighted for predicted functional significance by features including predicted TFBS motif creating/breaking effect and evolutionary conservation using pre-computed scores from the published FunSeq2 algorithm (http://funseq2.gersteinlab.org/downloads) with a higher score predicting higher likelihood of functional significance[43].

*Step 5: Calculate hotspot score.* Each hotspot is assigned a Hotspot Score as the product of the Donor Score (Step 3) and mean FunSeq2 score (Step 4) for all variants in the hotspot, to weigh both the number of variants and their predicted functional consequence in one metric. For example, in Fig. 1a, the purple box shows (Average FunSeq2 score)*(Donor Score) = 1.5*25 = 37.5

*Step 6: Calculate p-value for each hotspot in MRRs relative to the empirical null distribution (non-pMRR regions from Step 2).* For each hotspot score within pMRRs, we calculated a *p*-value by determining the proportion of null hotspots with hotspot scores greater than or equal to it. All *p*-values were adjusted for false discovery rate (FDR). Adjusted *p*-values equal to 0 are provided (Supplemental Data 2).

### Genomic analysis of hotspots
For all pMRRs, statistically significant hotspots (FDR adjusted *p*-value < 0.05, 707 hotspots), and top-scoring hotspots outside of pMRRs (top 707 null hotspots by Hotspot Score), we annotated regions using the ChIPSeeker function *annotatePeak*[91] (Fig. 1d). For HOMER motif analysis, we ran findMotifsGenome.pl on BED files of all pMRRs and statistically significant hotspots to identify known motifs (Supplemental Fig. 1a). For each variant within statistically significant hotspots, we made FASTA files with 20 bp sequences corresponding to either the WT or mutant sequence (variant at position 10). These were processed through HOMER using the findMotfs.pl function (Supplemental Fig. 1a). A BED file containing only the CDC20 promoter variants were processed through motifBreakR[92] using the known and discovered motif information from transcription factor ChIP-seq datasets in Encode[93].

To calculate the ETS motif distribution, we first made FASTA files containing 11 bp sequences corresponding to either the WT or mutant sequence (variant at position 6) from the 707 statistically significant hotspots with FDR-adjusted *p*-values < 0.05. If a sequence contained the GGAA motif, we counted how far each variant within a statistically significant hotspot occurred from the nearest GGAA (if more than one instance was detected). If the reverse complement, TTCC was identified, as the nearest ETS motif, we first rewrote the sequence as its reverse complement and then counted the distance. A consensus sequence was generated with Web Logo (https://weblogo.berkeley.edu/logo.cgi) using a re-oriented version of the 11 bp WT fasta file where the first G of the GGAA motif is always at position 5.

### Cohort comparison of Top 13 genes
We downloaded DESeq2-normalized read counts from GSE112509 for the Kunz cohort and quantile-normalized read counts from Firehose (Broad GDAC) for the TCGA-SKCM cohort. The Kunz cohort is made of 23 laser-microdissected melanocytic nevi and 57 primary melanomas[48]. The TCGA cohort consists of 81 primary and 367 metastatic melanomas[11].

For ICGC-MELA, we downloaded BAM outputs from STAR[94] from the European Genome-Phenome Archive (EGA) under Study ID EGAD00001003353. Gene counts were calculated using RSEM[95] and normalized by DESeq2[96]. This cohort consists of 56 melanomas from 46 donors and consists of 25 metastatic melanomas, 17 primary melanomas, and 14 cell lines derived from tumors[12].

For the Baggiolini cohort, we obtained raw counts from the supplementary material of the corresponding publication and normalized counts by DESeq2[96]. This cohort is made up of human pluripotent stem cell derived cells that are engineered to contain doxycycline-inducible BRAF$^{V600E}$. KO lines contain deletions to RB1, TP53, and P16. These cells were then differentiated into neural crest cells, melanoblasts, and melanocytes. For our study, we only considered WT and KO melanoblast samples that had activated BRAF$^{V600E}$ expression. In line with the corresponding publication, we consider KO melanoblasts to be melanoma-like (based on the ability to form tumors when subcutaneously injected into NSG mice) while WT melanoblasts were considered to be a non-tumorigenic precursor to melanocytes[49].

For each of the top 13 genes, we calculated the log$_2$ fold-change between metastatic and primary melanomas (TCGA-SKCM and ICGC-MELA), primary melanoma and nevi (Kunz), and KO and WT melanoblasts.

Kaplan–Meier curves of samples with and without the hotspot mutations were generated using the R packages *ggsurvfit* and *survival*[97]. Survival rates and corresponding *p*-values for high and low expressing tumors were downloaded from cBioPortal[98] (TCGA-SKCM) using the Onco Query Language (OQL): *GENE*: EXP < -0.5 and GENE: EXP > 0.5. Data was downloaded from cBioPortal.org and plotted with ggplot2.

### Cell culture
We obtained A375 (CRL-1619) and RPMI-7951 (HTB-66) cells from ATCC. SK-MEL-2, LOX-IMVI, SK-MEL-28, SK-MEL-5, UACC-62 cells were obtained directly from the NCI-

60 collection following written request and approval and were grown in RPMI-1640 media with 2 mM L-Glutamine (Gibco, 11875) with 10% FBS and 1X Pen/Strep. Newborn foreskin melanocytes were ordered from the specimen research core at the SPORE in Skin Cancer at Yale University. HEK-293FT cells were obtained from Invitrogen (#R70007). Cells were grown in a dedicated incubator set to 37°C at 5% CO2. A375 and HEK 293FT cell lines were grown in DMEM media (Corning, 10-013-CV) with 10% Fetal Bovine Serum (Gibco, 261470) and 1X Penicillin/Streptavidin (Pen/Strep, Sigma-Aldrich, P4333). Primary melanocytes were grown in OPTI-MEM (Gibco, 31985) containing 5% FBS, 1X Pen/Strep, 10 ng bFGF (ConnStem, F1004), 4 mL of 5 mM IBMX (Sigma, #I-5879), 1 ng/mL Heparin (Sigma, #3393), and 200 μL of 0.1 M dbcAMP (Sigma, #D-0627). SK-MEL-5 and RPMI-7951 were grown in EMEM media (Corning, 10-009-CV) with 10% Fetal Bovine Serum (Gibco, 261470) and 1X Penicillin/Streptavidin (Pen/Strep, Sigma-Aldrich, P4333). UACC-62, LOX-IMVI, UACC-257, SK-MEL-28, and SK-MEL-2 were grown in RPMI-1640 media (Corning, 10-040-CV) supplemented with 1X L-Glutamine (Gibco, # 25030081), 10% Fetal Bovine Serum (Gibco, 261470), and 1X Penicillin/Streptavidin (Pen/Strep, Sigma-Aldrich, P4333). Cells were grown in a dedicated incubator set to 37°C at 5% $CO_2$.

**Luciferase assays**. To make the luciferase vectors, we synthesized a 170 bp sequence containing the WT CDC20 promoter sequence (chr1:43,824,464-43,824,633) (GenScript). From this template, we amplified a 150 bp sequence using primers pGL3-CDC20_F and pGL3-CDC20_R (Phusion High-Fidelity PCR Master Mix, NEB M0531, Supplemental Data 12) that added restriction sites for SacI and XhoI to the 150 bp sequence. Both the pGL3-Basic Luciferase vector (Promega, E1751) and the CDC20 promoter amplicon were digested using SacI-HF (NEB, R3156S) and XhoI (NEB, R0146S) at 37°C overnight, followed by heat inactivation at 65°C for 20 min. Digested vector and amplicon were ligated using T4 DNA Ligase (NEB, M0202S) and transformed into OneShot Top10 Chemically Competent Cells (ThermoFisher, C404010). Individual colonies were mini-prepped and confirmed by Sanger Sequencing (Azenta).

Using the Q5 Site-Directed Mutagenesis kit (NEB, E0554), we induced variants in the WT sequence using primers designed by NEBaseChanger (https://nebasechanger.neb.com/, Supplemental Data 12). Sequences that were successfully mutated, as well as the WT pGL3-Basic vector and pRL-TK (Promega, E2241), were midi-prepped (Qiagen, 12941).

For all cell lines, 300,000 cells per well were seeded onto 6-well plates. All transfections were performed using 9 uL of Lipofectamine 2000 (Invitrogen, 11668), 1.5 μg of luciferase vector, and 1.0 μg of control pRL-TK (renilla), following the manufacturer's protocol. All transfections were performed at minimum in duplicate.

The following day, luciferase and renilla luminescence were measured using the Dual-Luciferase Reporter Assay System (Promega, E1910) per manufacturer specifications. Cells were lysed using 500 μL of 1X Passive Lysis Buffer and incubated for 15 min on an orbital shaker. 20 μL of lysate were added to clear-bottom 96-well plates. We ran three technical replicates per sample. Luminescence was measured on a GloMax 96 Microplate Luminometer (Promega) using a standard Dual Reporter Assay program. All luciferase values were normalized to renilla, as the internal transfection control. We then normalized all variant ratios to the corresponding average WT value. p-values were calculated using Student's t-test.

**Analysis of binding activity at the CDC20 promoter hotspot**. We used the encodeproject.org portal to find all TF ChIP-seq

assays targeting the following transcription factors: CREB1, ELF1, ELK1, ELK4, ETS1, ETV6, EZH2, GABPA, JUN, JUNB, JUND, MAX, MITF, MYC, SETDB1, SP1, and YY1. First, we downloaded only the processed BED files to identify the experiments that had signal at the CDC20 promoter hotspot. For these files, we then downloaded the corresponding bigWig files and merged files by transcription factor. We used deepTools to plot the signal profiles of each transcription factor[99].

To identify sequences that match the CDC20 promoter when mutated, we generated the following position weight matrices with the function seq2profile.pl from the HOMER algorithm[100]: CCGGAAGGCC (wild-type), CCAGAAGGCC (G525A), CCGGAAAGCC (G528A), CCGGAAGACC (G529A), and CCGGAAAACC (GG528AA). We scanned all promoters within pMRRs for instances of these motifs, down-sampled to 50,000 instances, and plotted the signal profile of each transcription factor at these locations using deepTools.

**GSEA of TCGA-SKCM, ICGC-MELA, and Kunz cohorts**. We classified each sample as CDC20-low or CDC20-high based on *CDC20* expression. Samples with less than the 25th percentile of *CDC20* expression were classified as low, while samples with greater than the 75th percentile of *CDC20* expression were classified as high.

We performed gene set enrichment analysis (GSEA) on the CDC20-low and CDC20-high samples using the Hoek et al. (2008) and Verfaillie et al. (2016) signatures (Supplemental Data 7). The following parameters were used: 1000 permutations, the phenotypes were always set as low versus high (ergo enrichment scores are positive for CDC20-low, negative for CDC20-high), permutations were performed on the gene set, the scoring scheme was set to weighted_p2, the metric used was tTest, and the normalization method was set to none.

**Analysis of scRNA-sequencing**. For the Wouters et al. (2020) dataset, we downloaded publicly available processed normalized read counts from GEO (GSE134432). Cell type classifications were taken directly from the corresponding manuscript[13]. A wilcoxon rank sum test was performed to calculate statistically significant changes in CDC20 expression between all cells classified as melanocytic and those classified as mesenchymal. For the Jerby-Arnon et al. (2018) dataset, we downloaded processed normalized read counts and tSNE cell annotations from the Broad Insitute Single Cell Portal[64] (SCP109). A wilcoxon rank sum test was performed to determine statistical significance between CDC20 expression of cells with low v. moderate and moderate v. high MITF signatures.

**Genome engineering of A375**. A375 cells were nucleofected on a Lonza 4D nucleofector according to manufacturer recommendations (P3 solution, nucleofection program EH-100). Each nucleofection was performed with $1 \times 10^5$ cells, 0.75 μL Cas9 Protein at 10 μg/μL (IDT v3 Cas9 protein, glycerol-free, # 10007806), and 0.75 μL of each sgRNA at 100uM (IDT) suspended in IDT Duplex Buffer (IDT, # 11-05-01-03) (Supplemental Data 12). Sham-nucleofections for WT A375 Cas9 controls were nucleofected with an equal volume of blank PBS. After nucleofection, cells were seeded into 500 μL of DMEM complete in a 24-well plate at standard incubator conditions.

72 hours post-nucleofection, cells were harvested, and split into 6-well culture for expansion and into lysis buffer for DNA extraction (homemade by GESC, formulation identical to Lucigen Quick-Extract buffer). PCRs were performed with Platinum Superfi II 2x master mix (Thermofisher, #12368010) and primers

against the sgRNAs target site (Supplemental Data 12). PCR products were sequenced by NGS using Illumina.

After confirmation of cutting activity at, the pools were single-cell sorted using a Sony SH800 cell sorter at 1 cell per well into 4 x 96-well plates with 100uL of DMEM, with 50% conditioned media, 5 μM Rock Inhibitor, and 100 μM sodium pyruvate. Plates were allowed to grow for ~10 days, then clones were harvested and re-screened using PCR primers against the targeted locus (Supplemental Data 12). Homozygous knockout clones were identified based on the presence of deletion junction and absence of the target locus. WT A375 Cas9 controls were sequenced at all gRNA target sites to confirm wild-type genotype. Homozygous knockout clones and wild-type Cas9 control clones were expanded, checked by STR profiling, tested for mycoplasma contamination, and used for subsequent experiments.

**Cell viability assay of A375 CDC20 promoter knock-outs and controls**. For each strain (A3, A10, and the wild-type Cas9 control), we seeded 1500 cells per well in a clear-bottom 96-well plate (Corning, #3903) in DMEM media containing 10% fetal bovine serum and 1X Pennicilin/Streptavidin (DMEM complete), DMEM complete with 30 nM dabrafenib (Selleck Chemicals, S2807), or DMEM complete with 1% DMSO. To measure viability, we used CellTiterGlo (Promega, G7570) as per the manufacturer's protocol. Plates were read on a GloMax 96 Microplate Luminometer (Promega) using the standard CellTiterGlo program.

**Cell migration assay of A375 CDC20 promoter knock-outs and controls**. Scratch assays were performed by seeding 1 million cells per well in a 6-well plate in DMEM complete media. Using a P200 pipette, we scratched the plate at indicated positions. Cells were washed with 1X PBS and imaged on a Nikon Eclipse Ts2. Cells were then plated with DMEM media with 1% FBS and 1X Pen/Strep. On the following day, cells were washed with 1X PBS and imaged.

To perform the Boyden Chamber Assay, we diluted rat tail collagen type 1 (Corning, 354249) down to 2 mg/mL in cold media (DMEM supplemented with 1% FBS) and brought the pH down to 7.5 using NaOH. We added 20 μL of 2 mg/mL collagen to the upper chamber of a FlouroBlok 96-well insert system (Corning, 08-771-006) with 8 μm pores and allowed it to polymerize for 30 min at 37 °C. DMEM supplemented with 10% FBS was added to the lower chamber as a chemoattractant. We resuspended cells in 1 μM CellTracker Green CMFDA Dye (Invitrogen, C7025) in serum-free media and incubated for 20 min at 37 °C. Cells were washed with PBS and split with Trypsin-EDTA (0.05%, Corning, 25300054). 8000 dyed cells per well were seeded onto the upper chamber and promptly placed into a Spark multimode microplate reader (Tecan) at 37 °C under 5% $CO_2$ and humidified conditions. For 48 h fluorescence (Ex: 493 nm, Ex bandwidth: 7.5 nm, Em: 517, Em bandwidth 10 nm) values were captured every hour across various Z-stacks with excitation from the bottom to identify cells that had migrated through the FluoroBlok insert.

**RNA-sequencing of A375 CDC20 promoter knock-outs and controls**. 300,000 cells of the parental A375 (in duplicate), two WT CRISPR/Cas9 clones (one replicate each), A3 (in duplicate), and A10 (in duplicate) were seeded on a 6-well plate. On the following day, we isolated RNA using the Qiagen RNeasy Plus Mini Kit (Qiagen, 74134). Samples were submitted to the Genome Technology Access Center at the McDonnell Genome Institute at Washington University School of Medicine for library preparation and sequencing.

Total RNA integrity was determined using Agilent Bioanalyzer or 4200 Tapestation. Library preparation was performed with 5 to 10ug of total RNA with a Bioanalyzer RIN score greater than 8.0. Ribosomal RNA was removed by poly-A selection using Oligo-dT beads (mRNA Direct kit, Life Technologies). mRNA was then fragmented in reverse transcriptase buffer and heating to 94 degrees for 8 min. mRNA was reverse transcribed to yield cDNA using SuperScript III RT enzyme (Life Technologies, per manufacturer's instructions) and random hexamers. A second strand reaction was performed to yield ds-cDNA. cDNA was blunt ended, had an A base added to the 3' ends, and then had Illumina sequencing adapters ligated to the ends. Ligated fragments were then amplified for 12–15 cycles using primers incorporating unique dual index tags. Fragments were sequenced on an Illumina NovaSeq-6000 using paired end reads extending 150 bases. RNA-seq reads were then aligned and quantitated to the Ensembl release 101 primary assembly with an Illumina DRAGEN Bio-IT on-premise server running version 3.9.3–8 software.

Read counts were normalized using DESeq2, comparing WT to mutant strains[96]. Principal component analysis was performed using the *plotPCA* function in the DESeq2 package. The heatmap was generated with *pheatmap* using z-score normalized counts of the manually curated list of 20 neural crest transcription factors[101] with FDR-adjusted *p*-values < 0.1 (between WT and mutant samples).

Gene set enrichment analysis was performed as previously described using the 25th and 75th quantile to establish CDC20-low and CDC20-high expression groups, respectively. The Tsoi et al. (2018) signature is provided as Supplemental Data 3 in the corresponding manuscript[2013]. For simplification, we added genes within the undifferentiated/neural crest-like subtype into the undifferentiated subtype, the neural crest-like/transitory into the neural crest-like subtype, and the transitory-melanocytic into the transitory category.

**qPCR analysis**. 300,000 cells were seeded on a 6-well plate with the corresponding media. Once the cells reached 60-80% confluency, we transfected Silencer Select siRNAs (ThermoFisher Scientific, Supplemental Data 12) in triplicate using Lipofectamine RNAi-MAX using the manufacturer's recommendation. 24 hours later we isolated RNA using the Qiagen RNeasy Plus Mini Kit (Qiagen, 74134) and generated cDNA using the Super Script III First-Strand System (Invitrogen, 18080051). qPCR was performed in quadruplicate with iTaq Universal SYBR Green Supermix (BioRad, 1725120) using the recommended protocol and primers to the corresponding gene (Supplemental Data 12) on a BioRad CFX Connect Real Time System. Cycle threshold (Ct) scores were normalized to the average *GAPDH* expression across all twelve replicates (ΔCt) and transformed using the following equation: $2^{-\Delta Ct}$. *p*-values were calculated using Student's *t*-test.

**Aneuploidy analysis**. Karyotyping and analysis of the A375 CDC20 Promoter Indel cell lines and the WT counterpart were performed at the Cytogenetics and Molecular Pathology Laboratory at Washington University School of Medicine. The cytogenetic test/ karyotype analysis was performed to assess aneuploidy (gains and losses of whole chromosomes), structural changes (chromosomal translocations, inversions, segmental deletions and duplications). This assay involves growing of cells in appropriate culture medium, hypotonic treatment, fixing cells, staining cells with GTG banding and microscopic examination. Twenty cells are counted for enumerating the number of chromosomes in a metaphase spread. Three of these metaphase spreads are digitally processed to produce a detailed karyotype/karyogram to perform a detailed study (analysis) for variant counts and structural aberrations.

Analyzing a metaphase is defined by band-by-band comparison between chromosome pairs.

To analyze aneuploidy in the DepMap and TCGA-SKCM cohorts, we used the data explorer portal (https://depmap.org/portal/interactive/) and cBioPortal (http://www.cbioportal.org/) to download the aneuploid scores and normalized CDC20 expression for all melanoma samples. Linear regression was performed using the lm() function in R.

**Xenograft mouse model.** Athymic nude female mice, ages 8 weeks, weighing approximately 20 to 22 grams, were purchased from the Jackson Lab (strain # 002019). All animal experiments were performed in accordance with protocols approved by the Institutional Animal Care and Use Committees (Protocol Number: 22-0263), and were performed in accordance with the Animals in Research: Reporting In Vivo Experiments (ARRIVE) guidelines for the care and usage.

Xenografts of three A375 human melanoma cell lines: WT, A3, and A10 clones were established in a blinded fashion, five mice in each group. $1.0 \times 10^6$ cells in 0.1 mL of PBS and Matrigel (1:1) were injected subcutaneously (s.c.) into the bilateral flanks of mice, using a 28- needle gauge insulin syringe to establish 30 total tumors (5 mice X 2 flank tumors each X 3 cell lines). The tumor cells were observed to grow and formed tumors after implantation. Mice were monitored weekly until primary tumors were seen, and subsequently, monitored and measured every four days. The tumor growth curve is determined by measuring the tumor volume using the equation: Volume = $(L \times W^2)$ x 0.52. At the end of the experiment (day 42 from inoculation date), mice were euthanized and tumors were excised, weighed, and stored in 10% formalin, OCT blocks, and flash frozen. Lungs and livers were examined with no evidence of metastasis noted. All the animal experiments were performed in accordance with the Animal Research: Reporting In Vivo Experiments (ARRIVE) guidelines for the care and usage. Tumor growth curves were visualized, and p-values were generated through permutation testing using the compareGC function in the DRAP R package[102]. Using the pwrss.t.2means function from the pwrss R package[103], we calculated that our experiment had a statistical power of 0.77.

**Reporting summary.** Further information on research design is available in the Nature Portfolio Reporting Summary linked to this article.

## Data availability

All raw and processed sequencing data generated in this study have been submitted to the NCBI Gene Expression Omnibus (GEO; https://www.ncbi.nlm.nih.gov/geo/) under accession number GSE206639. The supplementary data contains a list of the datasets used to define putative melanoma regulatory regions, the most significant hotspots, p-values and fold-changes corresponding to all luciferase assay experiments, a list of the ENCODE accession numbers used for ChIP-seq experiments, all GSEA summary statistics, normalized read counts for the RNA-sequencing experiments, tumor measurements related to Fig. 5, and a list of all primers used in this study.

## Code availability

All code used in this manuscript is available upon request.

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

## Acknowledgements

We thank the Genome Engineering and Stem Cell Center (GESC) at Washington University in St Louis for their Cell Line Engineering Services of A375 cells, the Cytogenetics and Molecular Pathology Laboratory at Washington University School of Medicine for karyotyping analysis, and the Genome Technology Access Center at the McDonnell Genome Institute at Washington University School of Medicine for help with next generation sequencing and genomic analysis. The Genome Technology Access Center is partially supported by NCI Cancer Center Support Grant #P30 CA91842 to the Siteman Cancer Center from the National Center for Research Resources (NCRR), a component of the National Institutes of Health (NIH), and NIH Roadmap for Medical Research. This publication is solely the responsibility of the authors and does not necessarily represent the official view of NCRR or NIH. We thank Megan Glaeser and Rebecca Cunningham for critical reading of the manuscript, and Catie Newsom-Stewart for help with Fig. conceptualization and design. This research was supported by the Melanoma Research Alliance Young Investigator Award #566840. P.G. was supported by NSF DGE-1745038.

## Author contributions

P.M.G. and C.K.K. conceived the study. Mouse xenografts were performed by A.O. and J.L.M. in the laboratory of R.F., and the Boyden chamber assay was performed by V.A.M. in the laboratory of G.D.L. A.P.Z. and P.M.G. performed the luciferase assays. P.M.G. performed all other laboratory experiments and all data analysis. P.M.G. generated figures and wrote the manuscript. C.K.K. commented on the manuscript. All authors contributed to editing the manuscript. All authors have read and approved the manuscript.

## Competing interests

The authors declare no competing interest.

## Additional information

**Peer review information** This manuscript has been previously reviewed at another Nature Portfolio journal. The manuscript was considered suitable for publication without further review at *Communications Biology*.

