## [Peer Review File · Communications Biology]

Reviewer #1 (Remarks to the Author):

1. Question 5 was only partially addressed. ELF1 ChIP-qPCR wasn't done on cell lines with wt or CDC20 promoter mutations. Authors should engineer the mutations, and conduct ELF1 ChIP, together with H3K27ac, H3K4me3 at the CDC20 promoter to study the true epigenetic effects of these mutations in the endogenous locus. This would perhaps explain the discrepancy between their results and those in He et al, reporting that mutations lead to a modest increase in CDC20 expression.

To clarify:

- (1) We have already previously engineered mutations in a relevant human melanoma cell line A375 (Fig 4A). Our study engineered a small (7 bp) and large (75 bp) indel in independent clones. We certainly acknowledge that these are not the exact mutations, but as noted in our initial response, extensive attempts to perform prime editing were unsuccessful in our hands, as we understand is not atypical for some loci with the still developing prime editing technique. Importantly, previous studies have shown congruent results between single nucleotide variant engineering and small indels¹, further supporting the use of small indels as a reasonable approach.
- (2) Of note, we performed knockdown of the candidate ETS factor ELF1 and note decreases in endogenous CDC20 expression, while knockdown of another ETS factor GABPA did not alter CDC20 expression, indicating specificity of the ELF1 effect (Supp Fig 4).

We surmise that the suggested ChIP-Seq/PCR may be echoed from the He et al study, which notably was done for the endogenous wild-type CDC20 promoter (no mutations) in the presence of an overexpressed flag-tagged ELK4 in HEK293 cells (based on the figure labeling), which is not a melanoma cell line. Moreover, despite being bound by ELK4 in this overexpression setting, there was no change in luciferase activity from reporters in the non-melanoma HEK293 when ELK4 was knocked down, highlighting the importance of cellular context (melanoma cell line) for such assays. With this specific case and more generally, we anticipate limited added value in demonstrating binding of ELF1 in an overexpression, tagged protein context.

- (3) It is important to clarify that He et al reported a modest increase in reporter luciferase activity, not CDC20 expression. We did see an increase reporter activity with GG58AA, most notably in HEK 293 cells, matching the He et al study² (Fig 2B). Our study shows a decrease in CDC20 expression with engineered mutations in the genome of a melanoma cell line and a decrease in luciferase reporter activity in 7 melanoma cell lines, 1 primary melanocyte line, and HEK293 (non-melanoma cell line); these additional luciferase assays were performed due to the reviewer's initial request and strengthened our conclusions that the CDC20 promoter variants lead to down-regulation of luciferase reporter activity. Also, as noted in our initial response, the promoter fragment used in the He et al luciferase assay was substantially longer and extended into an annotated UTR, likely confounding direct comparisons between studies.

Overall, engineering the precise variants is a technically challenging (potentially not achievable with current techniques), highly costly (in time and monetary resources) undertaking, and without the ability to reasonably test such variants in a *de novo* tumor model where the effects could be assayed across the tumor lifespan, we would argue the added value is at most

incremental beyond what we have reported. While the reviewer requests additional ChIP-PCR experiments on ELF1, H3K27Ac, and H3K4me3, we have chosen to focus our current study on the broad transcriptional changes caused by the variant altering gene expression rather than changes in epigenetic marks induced by the variant. We agree that transcriptional regulation/epigenetic modifications are important to understand and should be thoroughly investigated in future studies.

2. Question number 7 was only partially addressed. The neural crest state is well described in the literature and there are plenty of datasets to verify the anti-correlation between CDC20 and Neural-crest state. Therefore, authors should check at the single cell level (e.g., Rambow et al, Wouters et al, Karras et al, Tirosh, Jerby-Arnon et al) the expression of CDC20 within different phenotypic states. Bulk analysis using Hoek and Verfaillie is a good starting point which however is limited by the loss of intratumor heterogeneity intrinsic to bulk RNA-seq. Moreover, SOX10 ChIP-seq is available for melanoma cell lines and authors should verify the binding of SOX10 to the CDC20 promoter region.

We thank the reviewer for this suggestion. We have explored the scRNA-sequencing datasets suggested. While identifying an anti-correlation on a single-cell basis between SOX10 and CDC20 would lend support to our hypothesis, low read counts and sparse datasets generally yielded insignificant and/or low correlation coefficients. Therefore, we sought to determine alternative solutions when applicable.

- (1) Wouters et al – In this study, only 3% (n = 133) of cells had log-normalized read counts greater than 2 for both SOX10 and CDC20. When only considering these cells, we calculated a Pearson's correlation coefficient of 0.16 with p-value = 0.07. Alternatively, we used the available data and cluster annotations to perform an analysis similar to Figure 1C in the Wouters et al publication. The 10 samples that were scRNA-sequenced were categorized as either melanocytic or mesenchymal based on various gene markers and gene sets, which correlated almost always with SOX10 (melanocytic) and SOX9 (mesenchymal) expression. We therefore looked at the distribution of CDC20 counts in each cluster and note a statistically significant decrease (p-value < 2.2×10^{-16}) between CDC20 expression in melanocytic clusters, which have relatively higher levels of SOX10 (Fig 1A in Wouters et al.), and those categorized as mesenchymal (high SOX9) (presented in a new Figure 3C). This observation further supports our hypothesis that lower levels of CDC20 may be preferential in supporting a melanocytic/proliferative state. The reviewer also brings up an interesting notion in whether SOX10 regulates CDC20 expression. We did not observe any binding activity of SOX10 at the CDC20 promoter, suggesting that SOX10 does not indirectly regulate itself by regulating CDC20.
- (2) Hodis et al and Karras et al – We were not able to analyze these studies as <1% of the cells in this study had log-normalized reads greater than 2 for both SOX10 and CDC20.
- (3) Rambow et al – 10% of cells (n = 74) had log-normalized read counts greater than 2 for CDC20 and SOX10. This yielded an insignificant Pearson's correlation coefficient of 0.18 (p = 0.12).
- (4) Tirosh/Jerby-Arnon – The Jerby-Arnon study performed scRNA-seq on several malignant melanomas and melanoma cell lines. 21% of cells (n = 844) had log-normalized read counts greater than 2. This yielded a statistically significant positive correlation between CDC20 and SOX10 in IGR37 and UACC-257 but an insignificant

correlation within the malignant melanoma samples. We note that many cells (54% for IGR37 and 47% of UACC257) had either greater than 2 log-normalized read counts for CDC20 and zero for SOX10 or vice versa, suggesting that there are many data points we cannot account for due to sparsity. This study also classified cells using the MITF signature from the Tirosh et al. paper as having either high, moderate, or low levels of this signature. As expected, we observed a statistically significant decrease in CDC20 expression between the low, moderate, and high MITF categories in IGR37 and UACC257. We have incorporated this analysis into our manuscript (new Figure 3D and 3E).

Overall, we identified a clear trend between CDC20 expression and a melanocytic/proliferative and mesenchymal/invasive state across multiple datasets.

3. Question number 11 lacks a critical *in vivo* loss of function experiment using shCDC20 or CRISPR/cas9/sgCDC20 melanoma cells vs corresponding shNTCs or shCDC20 controls, to assess how levels of CDC20 impact primary tumor formation and metastatic progression.

Per the initial reviews (“The study is missing an *in vivo* analysis of A375wt vs. clones A3 and A10, or shCTRL vs shCDC20 to assess how these mutations and the levels of CDC20 impact primary tumor formation and metastatic progression.”), we performed *in vivo* xenograft experiments with the engineered A3/A10 and WT A375 lines (Fig. 5), excitingly revealing a growth advantage *in vivo* for the greater decrease of CDC20 present in the A3 clone. We also did not see metastasis in any of the xenografts, WT or mutant cell lines. While we agree that, in principle, a genetic analysis of the role of CDC20 in primary tumor formation would be very interesting, the complexity and time needed for the required genetics is beyond the scope of this study (e.g. complete CDC20 deletion is lethal; tissue specific loss of function in melanocytes using Cre/Lox would almost certainly be lethal for the melanocytes precluding analysis of tumor onset rates in the various genetically engineered mouse melanoma Cre/Lox models; tissue-specific *in vivo* partial knock-down of CDC20 to assess primary tumor formation requires time and resource intensive generation of novel inducible hypomorph/knockdown strains and breeding into melanoma GEMs and primary tumor/metastasis analysis).

4. Moreover, the paper falls short at addressing the main unanswered question: how levels of a cell division protein can impact melanoma phenotypic state.

We agree that the mechanism remains incompletely answered in this initial report. However, we feel that identifying a correlation between CDC20 expression and a particular phenotypic state, which may in a subset of samples be driven by a functional non-coding promoter variant, will be of general interest and will drive further inquiry. We could speculate that alterations in subtle aspects of cell cycle progression may be linked to transcription/gene regulatory networks directing phenotypic state as has been seen in ES cells^{3,4}. Further, one could speculate about non-canonical roles for the larger APC/C-CDC20 containing complex using its known E3 ligase function to target a wider array of proteins beyond the Cdks, perhaps even transcription factors^{5,6} (e.g. E2F1). Such interesting possibilities, and likely others, will require future analysis beyond the scope of this study, as included in Discussion (pgs. 31-32).

Reviewer #2 (Remarks to the Author): The authors have added additional experiments showing the complexity of noncoding promoter mutations in impacting gene expression. New analyses of TFs support a regulatory role of the CDC20 promoter mutation. I do not find convincing support for an evolutionary (negative) selection of CDC20 promoter mutations at relapse and this part of the story should be revised to reflect the speculative aspect.

2.1: I maintain that the expression comparison between primary and metastases is odd and does not make logical sense, given the aim of the study to search for noncoding (promoter) mutations. No comment is made on the fact that 6 out of the top 8 promoter genes are ribosomal genes, all associated with down-regulation (RPS3A, RPS27, RPS20, RPS14, BPL18A, RPL13A). These could also be interesting candidates to follow up on separately, or perhaps there is a mapping artefact in largely homologous promoter DNA sequences between ribosomal genes?

We understand that an ideal analysis could aim to identify eQTLs or use allele-specific expression as decision criteria. Unfortunately, of the 200 melanomas that have been whole genome sequenced to adequate read depth (>30X), the minority (~50) have also been analyzed by RNA-sequencing which renders eQTL and ASE analysis underpowered for most variants. However, we have amended the manuscript to include expression differences between WT and mutant samples for the top 13 hotspots (new Supplemental Figure 2, pgs. 5-6), which is described in further detail in the next comment (Point 2.2).

We also understand the reviewer's concern regarding using \log_2 fold-changes between primary and metastatic samples as a decision criterion. Instead of filtering out promoter hotspots due to variant-specific criteria (like expression changes between WT and mutant samples), we filtered out the genes that were likely to be targeted by the variants (based on proximity) which had no or little change in gene expression, as this may indicate no function in cancer initiation/progression/metastasis. Although we would have preferred to compare expression between normal and cancerous tissues, primary and metastatic melanoma samples were used in the ICGC-MELA and the TCGA-SKCM cohorts as these were the only available samples; samples from normal tissues were not RNA-sequenced in these studies. However, in another three cohorts, we compared gene expression between non-cancerous and cancerous samples (e.g. nevi and melanoma from Kunz cohort). We recognize this criterion relies on several assumptions and amend the manuscript to clearly state the limitations of our criteria.

We agree with the reviewer regarding the remarkable presence of ribosomal genes in the top candidates. The RPS27 promoter hotspot has been validated in a previous study⁷. To our knowledge there is no mapping artefact in the promoters of these genes. Instead, we suspect these promoters are susceptible to UV damage due to high binding activity of ETS transcription factors⁸ and CTCF⁹. Promoters with the motif CTTCCG, which include RPL13A, RPL18A, TERT, CDC20, and DPH3, have been previously identified to be recurrently mutated in melanoma¹⁰.

In conclusion, we agree with the reviewer's concern regarding the suboptimal decision criteria used and have edited the manuscript to clearly state our rationale in using the decision criteria in the manuscript and the limitations of these criteria (pgs. 5-6, 31-32).

2.2: The average expression change of promoter mutated vs ctrl should be displayed for all the candidates. This might motivate a more objective criterion to leave out the ribosomal genes for further analysis.

We have added these plots as a new Supplemental Figure 2 and made corresponding remarks in the manuscript (pgs. 5-6). We observed significant changes in gene expression dependent on promoter status for CANX, CDC20, TERT, DPH3, PES1, RPS27, SLC30A6, RPL13A, and HNRNPUL1. We do recognize that their functional significance is yet to be determined and may prove to be biologically of importance in specific contexts, as we found in the case of CDC20, and now more clearly state this in the manuscript (pgs. 5, 6, 31).

2.3: Ok. It is important to further stress in the manuscript that the actually observed promoter mutation(s) are not generated - especially in light of the variable phenotype of the different mutations in different cell line contexts.

We have amended the manuscript to clearly state these details.

2.5: The authors should clarify in Fig S3DE. I assume these are the proportion of genome-wide UV signature mutations in patient samples with and without promoter mutations. What is the interpretation? It does not necessarily mean that the promoter mutations themselves are generated by UV signature - as is interpreted by the authors. Annotating specific mutations to SBS signatures is a tricky analysis and can only be done using probabilities.

We agree that we cannot solely state that promoter mutations are caused by UV damage based on this data and have amended the manuscript to avoid this language (pg. 12). We also further emphasized in the discussion previous investigations into the source of promoter mutations.

2.6 and 2.7: The updated analysis and conclusions p18 are fairly weak (also related to my next comment) and only suggestive. In the lack of stronger evidence, this section should be significantly reduced and toned down. I raised copy number as a concern for VAF evaluation, and it appears that the authors have slightly misunderstood this. The main useful information is the copy number level of the CDC20 gene itself (this can easily be computed) - not genome-wide copy number changes.

Previous comment: VAF distributions for BRAF, NRAS, and TERT appear unusually broad (Fig 3). Driver mutations in these genes are usually clonal, having a VAF close to 0.5. Moreover, VAF can be deceiving as tumor purity is not subtracted (as well as copy number level). CCF is the right metric for comparative clonality estimates.

We appreciate this comment from the reviewer and have largely removed this section from the manuscript. We further seek to address this comment in concert with response to 2.12.

2.12: That a mosaic of different TME and cancer cell fractions are present in cancers is now well established and certainly not surprising, but also does not support the proposed CDC20 primary-vs-metastasis link. Without further support, I remain unconvinced that there is a selection of promoter WT tumours at relapse in the same patient.

We agree that our updated analysis is only suggestive and, with available WGS and samples, lacks sufficient statistical rigor thus far. We therefore remove insinuations of negative selection of cells with the promoter WT mutations. To partially address the reviewer's concern about the copy number level of CDC20, we note that there were no detected copy number alterations of

CDC20 in the ICGC cohort; only 3/287 samples in the TCGA cohort had amplifications for the CDC20 gene.

REFERENCES

1. Yanchus, C. *et al.* A noncoding single-nucleotide polymorphism at 8q24 drives IDH1-mutant glioma formation. *Science* 378, 68–78 (2022).
2. He, Z. *et al.* Pan-cancer noncoding genomic analysis identifies functional CDC20 promoter mutation hotspots. *IScience* 24, 102285 (2021).
3. Oh, E. *et al.* Gene expression and cell identity controlled by anaphase-promoting complex. *Nature* 579, 136–140 (2020).
4. Gonzales, K. A. U. *et al.* Deterministic Restriction on Pluripotent State Dissolution by Cell-Cycle Pathways. *Cell* 162, 564–579 (2015).
5. Peart, M. J. *et al.* APC/CCdc20 targets E2F1 for degradation in prometaphase. *Cell Cycle* 9, 3956–3964 (2010).
6. Wang, W., Wu, T. & Kirschner, M. W. The master cell cycle regulator APC-Cdc20 regulates ciliary length and disassembly of the primary cilium. *eLife* 3, e03083 (2014).
7. Floristán, A. *et al.* Functional analysis of RPS27 mutations and expression in melanoma. *Pigm Cell Melanoma R* 33, 466–479 (2020).
8. Mao, P. *et al.* ETS transcription factors induce a unique UV damage signature that drives recurrent mutagenesis in melanoma. *Nat Commun* 9, 2626 (2018).
9. Poulos, R. C. *et al.* Functional Mutations Form at CTCF-Cohesin Binding Sites in Melanoma Due to Uneven Nucleotide Excision Repair across the Motif. *Cell Reports* 17, 2865–2872 (2016).
10. Fredriksson, N. J. *et al.* Recurrent promoter mutations in melanoma are defined by an extended context-specific mutational signature. *Plos Genet* 13, e1006773 (2017).